# The dawn of biophysical representations in computational immunology

Eric Wilson[1], Akshansh Kaushik[2], Soumya Dutta[3] and Abhishek Singharoy[3]

[1]Department of Immunology and Immunotherapy, Icahn School of Medicine at Mount Sinai, New York, NY, USA; [2]School of Molecular Sciences, Arizona State University, Tempe, AZ, USA and [3]Biodesign Institute, Center for Applied Structural Discovery

## Perspective

Computational biophysics; Molecular; immunological, and structural biology; Bioinformatics; Vaccine development

**Corresponding author:**
Abhishek Singharoy;
Email: asinghar@asu.edu

A.K and S.D have contributed equally to this work.

### Abstract

Computational immunology has been the breeding ground of some of the best bioinformatics work of the day. By melding diverse data types, these approaches have been successful in associating genotypes with phenotypes. However, the representations (or spaces) in which these associations are mapped have primarily been constructed from some omics-oriented sequence data typically derived from high-throughput experiments. In this perspective, we highlight the importance of biophysical representations for performing the genotype–phenotype map. We contend that using biophysical representations reduces the dimensionality of a search problem, dramatically expedites the algorithm, and more importantly, offers physical interpretability to the classes of clustered sequences across different layers of complexity – molecular, cellular, or macro-level. Such biophysical interpretations offer a firm basis for the future of bioengineering and cell-based therapies.

## Introduction

The core responsibility of our immune system is to protect the body from pathogens and cancers. The need to target and activate the immune system reproducibly has been underscored by the recent pandemic and the rise of anticancer therapies that rely on immunological mechanisms. This has generated a focused enthusiasm for gaining a detailed description of the immune system. However, the human immune system is incredibly complex, and often regarded as one of the most challenging topics in biology. The sheer size of sequence and population diversity in proteins associated with the immune system presents a formidable obstacle to mapping their network of interactions within a tractable space. For instance, T-cell recognition of antigens is driven by human leukocyte antigens (HLA) genes encoding Major Histocompatibility Complexes (or MHCs), which are among the most polymorphic germline genes in the human genome that contain tens of thousands of variants across populations (Barker *et al.*, 2023). Moreover, the somatic hypermutations involved in the function of T-cell and B-cell receptors make them the most polymorphic human proteins in known existence, with theoretical estimates of T-cell receptor (or TCR) diversity reaching over $10^{61}$ potential sequences. A more conservative estimate places TCR diversity in the range of $10^7$ receptors (Mora and Walczak, 2018), which still offers an incredibly vast range of human variations. The desire to account for this diversity and predict its associated non-linear relationships has motivated the genesis of the field of computational immunology to develop methods to analyze and predict immune outcomes based on this data (Figure 1). Computational immunology has transformed our understanding of the immune system by enabling the integration of massive amounts of biochemical and biological data. Simple mathematical models to study disease transmission can be traced to the early 20th century (Ross, 1911; Brauer, 2017). By leveraging population data, it clarified the relationship between the size of mosquito populations and malaria incidence, which led to improved malaria control. The power of computational immunology expanded significantly in the information age with the advent of high-throughput sequencing, proteomics, and the growing availability of experimental and clinical data further empowered by advances in computational technology. These advancements have enabled computational techniques to tackle more complex immunological questions. Consequently, computational immunology has now been applied to a broad spectrum of applications including vaccine design (He and Zhu, 2015), predicting population-level mortality rates (Wilson *et al.*, 2021), and forecasting the outcomes of immune checkpoint blockade therapies (Chowell *et al.*, 2018).

Due to the availability and ease of collection of protein and amino acid sequence information, most computational immunology approaches primarily rely on sequence data for their predictions (Ansari and Raghava, 2010; Jespersen *et al.*, 2017; Peters *et al.*, 2020). However, recent advances in machine learning and protein modeling have caused an explosion in the synergistic incorporation of biophysical information and modeling into existing computational immunology approaches (Andersen *et al.*, 2006; Wilson *et al.*, 2024). Such integrations have already



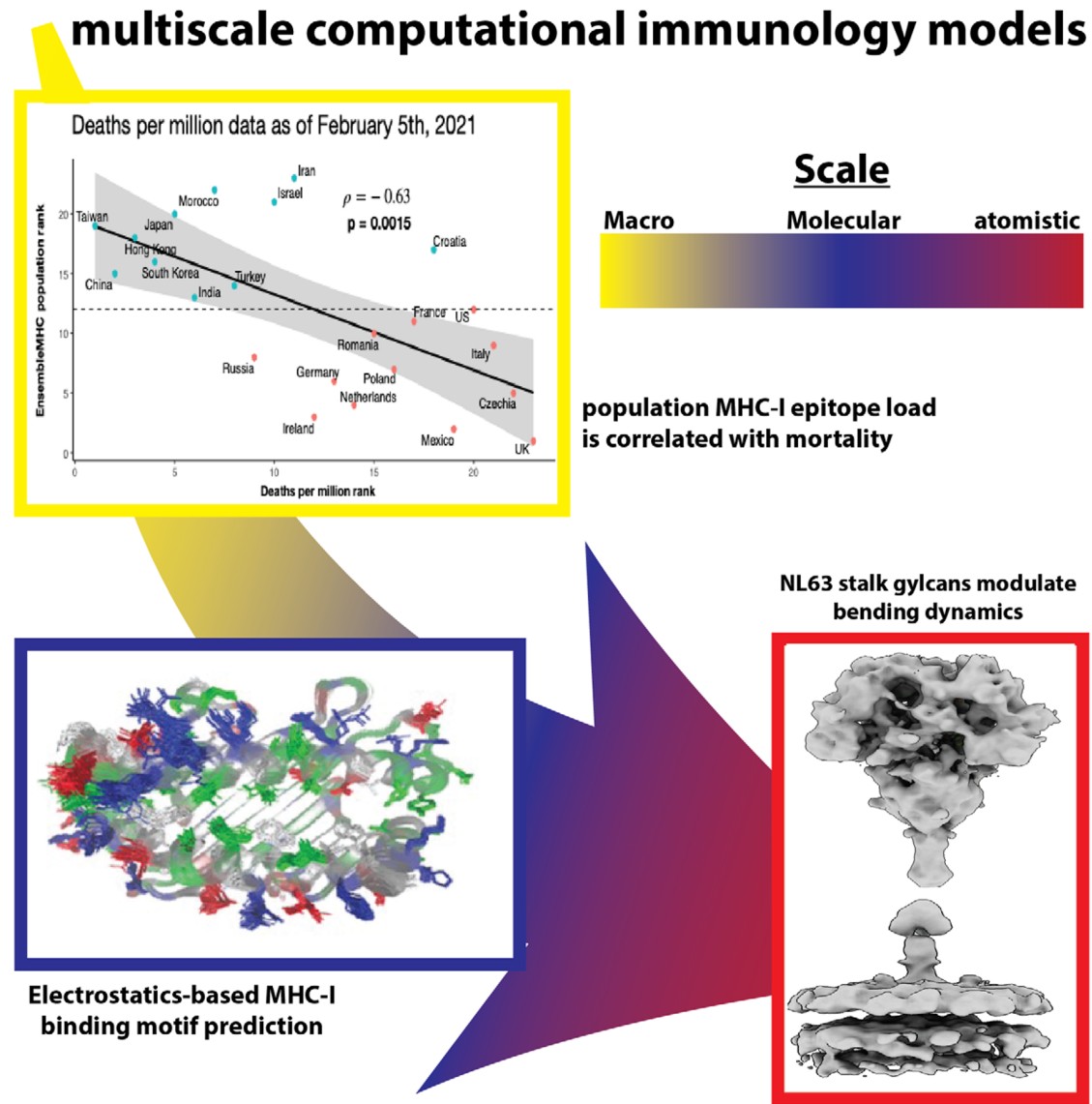

**Figure 1.** The scales of computational immunology models from atomistic to macroscales.

shown a profound improvement in the accuracy of models, but also enable novel insights into previously inscrutable mechanisms, advancing computational immunology models into the next era. In the following perspective, we will explore immune-related models ranging from atomistic environments to macro-level systems, demonstrating how biophysics can be used to enhance predictive accuracy and improve our overall understanding of immune responses.

## A perspective on biophysical models

Computational immunology has been dominated by bioinformatics, primarily due to a push from recent findings in genomic and proteomic technologies that compose around 31 different databases today (Rigden and Fernández, 2023). Historically, it allows the study of complex protein–protein interactions across a diversity of sequences (Petrovsky and Brusic, 2002). Recently, deep learning approaches have offered rapid access to molecular structures from sequences (Jumper *et al.*, 2021), which has extended

the realm of bioinformatics to structure-guided models of immune interactions (Bradley, 2023). However, the physical formulation of intermolecular interactions is statistical, which entails an ensemble description of conformations that remains obscure in the bioinformatics approaches. These ensembles capture transition in the order–disorder transition of the molecules, flexibility, and thermal effects, as well as solvation and microenvironmental impacts on structure. Attempts to overcome such limitations of traditional computational immunology open the doors for employing biophysical tools to take MHC, TCRs, and antibody predictions beyond the sequence-only or sequence-structure paradigm (Raha *et al.*, 2022; Deng *et al.*, 2023; Demerdash and Smith, 2024). Notwithstanding the computationally expensive biophysical simulations, it generates unique representations and metrics that connect collective molecular properties with phenotypic and even population outcomes. We break down the biophysical advances in the realm of atomistic, molecular, whole-cell, and macro-level modeling, and highlight how biophysical entities of Figure 1 are acting or can be leveraged as novel representations for learning in computational immunology, as

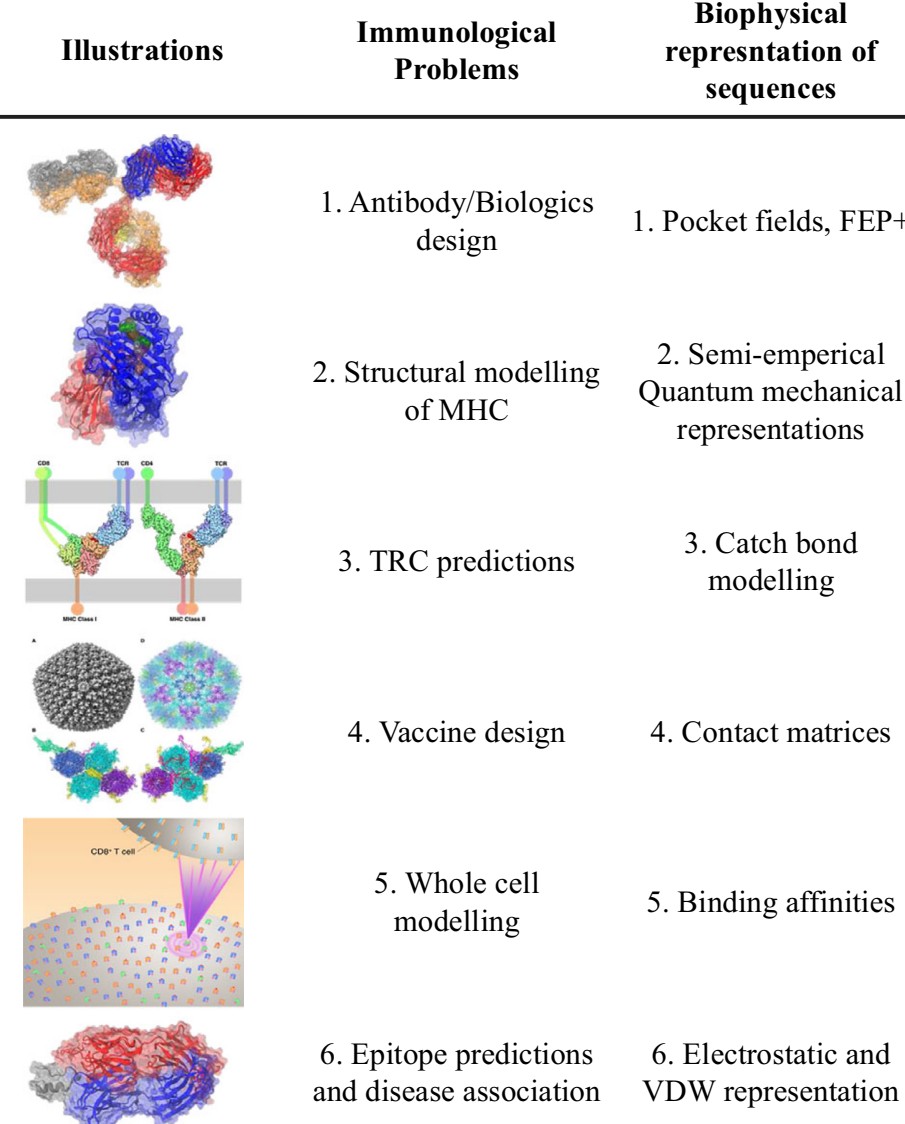

| Illustrations | Immunological Problems | Biophysical represnation of sequences |
|---|---|---|
| | 1. Antibody/Biologics design | 1. Pocket fields, FEP+ |
| | 2. Structural modelling of MHC | 2. Semi-emperical Quantum mechanical representations |
| | 3. TRC predictions | 3. Catch bond modelling |
| | 4. Vaccine design | 4. Contact matrices |
| | 5. Whole cell modelling | 5. Binding affinities |
| | 6. Epitope predictions and disease association | 6. Electrostatic and VDW representation |

**Figure 2.** A comprehensive list of immunological problems and their biophysical representations. Illustrations – 1. Antibody (PDB-1IGT), 2. MHC (PDB-1HHK), 3. TCR (from RCSB-PDB), 4. Viral vector ChAdOx1, 5. Whole-cell illustration, and 6. Epitope (PDB-3PP4).

complements to the traditional sequence or structural methods (Figure 2).

### Atomistic description

We start with biophysical descriptors in computational biology arising from detailed interactions of antibodies, MHCs, and TCRs.

### Free energy description of antibodies

Since the first antibody structure was deposited in 1976, the number of antibody structures in the Protein Data Bank (PDB) has grown, and it now represents approximately 2.1% of the total entries (Ferdous and Martin, 2018). Many computational tools now use only the antibody data, as opposed to general protein data, due to the increased performance (Ponomarenko and Bourne, 2007; Młokosiewicz et al., 2022). To this end, the Structural Antibody Database (or SAbDab) collects, curates, and presents an ensemble of antibody structures from the PDB (Schneider et al., 2022). Such databases allow for the

prediction of the affinity of antibody–antigen interfaces by combining the biophysics of protein–protein interactions with deep learning approaches (Hummer et al., 2023). In fact, a significant improvement in the ranking and prediction of affinity predictions is observed by combining all-atom free energy methods like Free energy perturbation or FEP+ with focused machine learning approaches like QuanSA (Cleves and Jain, 2018). Using such a combination of biophysics and informatics, the affinity of the CR3022 antibody is optimized to the spike protein of the SARS-CoV-2 Omicron strain, achieving a high success rate with up to a 17-fold affinity increase (Cai et al., 2024). Going beyond simple geometric 3D coordinate representations of ligands (Cleves and Jain, 2018), a novel metric of multiple-ligand alignment is employed using so-called pocket fields to learn affinities. Unlike the learning of real geometries that are quite high-dimensional, the learning of smoother functions like the 3D fields (with known map to the SMILE or peptide sequences) offers learning across a broad diversity of molecular identity and conformation, without overfitting the loss function. In conclusion, the

application of free energy-augmented antibody design underscores the growing power of biophysical modeling to not only understand but also engineer biological systems for specific therapeutic outcomes.

## Structural modeling of MHC (Major Histocompatibility Complex)

In 1968, Snell examined the concept of transplantation and came across the term histocompatibility polymorphism (Hull, 1970; Garrido, 2024). MHC proteins play a crucial role in immune mechanisms due to their involvement in activating T cells and B cells (Janeway *et al.*, 2001; Wieczorek *et al.*, 2017). Structural modeling of these complexes offers insights into the mechanism of the several pathways relevant to immunogenicity (Keller *et al.*, 2022). The MHC protein is one of the most polymorphic proteins in humans (Barker *et al.*, 2023), but despite the high polymorphism, the structure of the MHC binding groove is highly conserved (Wilson *et al.*, 2024). Researchers found that the second and last residues are key anchors for peptide binding to the MHC class-I binding groove (Janeway *et al.*, 2001), a discovery made through X-ray diffraction studies (Zhang *et al.*, 1998). Since countless peptides can bind to MHC, many generated by frameshift events, and lack evolutionary context for multi-sequence alignments, crystallizing all polymorphic complexes is unfeasible. A biophysical approach is thus needed to model MHC–peptide complexes for further study.

Conventionally, there are three ways to model structures: molecular dynamics, molecular docking, and homology modeling (Bertoline *et al.*, 2023). The unifying protocol to design a model for MHC is as follows: the first part is to generate a peptide conformation using a PDB template, the second step involves docking of the peptide, and finally optimizing the overall structure. Multiple sources are available to model MHC-I complexes such as Dock-Tope, GradDock, APE-Gen, AlphaFold2, and RoseTTAfold (Rigo *et al.*, 2015; Kyeong *et al.*, 2018; Abella *et al.*, 2019; Bryant *et al.*, 2022). Although these methods are highly accurate, some of them are highly computationally heavy or applicable only to the MHC class-I molecule due to the heterodimeric binding pocket observed in MHC class-II molecules. Recently, a state-of-the-art method, *PANDORA*, shows potential to design even MHC class-II molecules, and also offers some tunability while modeling. Its energy-based definition of loop conformations is shown to outperform most of the methods previously introduced in terms of accuracy and computational efficiency (Parizi *et al.*, 2023). However, there still is a need for a tool that models complex structures by capturing the biophysical attributes of the peptide–MHC complex instead of exploiting sequence similarity and templates. Large datasets to benchmark biophysical properties across a range of MHC systems – similar to MISATO (for MD simulations of 20,000 protein-ligand systems) or 100-protein NMR spectra (for protein dynamics) – do not yet exist in this space. A very promising result is that semi-empirical quantum mechanical representations can now be embedded in these data sets to refine the associated protein structures. Once similar datasets start existing for the broad class of MHC proteins, such quantum chemistry representations can likely be extended to the peptide–MHC predictions, for example, with PANDORA or other tools. Ultimately, improvement to MHC modeling and subsequent extraction of generalizable biophysical properties will lead to better predictions of immunogenicity. Highlighting this point, a thorough structural study demonstrated that a non-anchor position mutation in an MHC-I peptide, presented by an ovarian cancer tumor, modified both the structural and dynamic properties of the bound complex. These changes resulted in optimal confirmations for interaction and subsequent activation of cognate T cells (Devlin *et al.*, 2020). Such an observation would be difficult, if not impossible to determine from sequence alone and emphasizes the value of structural considerations when studying immunogenicity.

## Catch bond description of TCRs

Catch bonds have been referred to as the interaction between various biomolecules and biomolecular surfaces, where the lifetime of the bond increases with the application of tensile force on the bond (Marshall *et al.*, 2003; Hertig and Vogel, 2012). The atomistic detail of catch bond formation had remained elusive for a long period of time, but the general explanation was given by a two-state model or a two-pathway model. In the two-state model, the receptor-ligand complex is theorized to exist in two distinct states, a short-lived and a long-lived state. The application of force loosens the interaction between the binding site and a regulatory site, which drives the whole complex toward the long-lifetime state (Hertig and Vogel, 2012). In the two-pathway model, the receptor-ligand complex undergoes unbinding via two distinct pathways with different $K_{off}$ values, and the application of tensile force triggers the allosteric change that leads the unbinding to happen via the pathway with a high energy barrier, thereby the long-lifetime (Sokurenko *et al.*, 2008). Such catch bonding has been observed at the TCR-peptide–MHC immune synapse, and more importantly, immunogenicity has been attributed to the strength of the catch bond formation (Choi *et al.*, 2023). Hence, catch bonds offer a biophysical descriptor of MHC alleles for presenting peptides to the TCRs. Interestingly, unlike binding affinity, catch bonds uniquely capture the system's out-of-equilibrium properties. Therefore, it can capture the state of the immune synapse under stress, which rectifies the frozen stationary picture of complexes drawn by the affinity measures. This descriptor is computable using Steered MD simulations (Schoeler *et al.*, 2014) and more recently using metadynamics methodologies (Ccoa and Hocky, 2022), offering insights into how sequence changes reflect in non-equilibrium interaction changes. However, both the experimental and computational biophysical methods for tracking catch bonds are resource-intensive, so high-throughput measurements are yet missing, in turn impacting the extensive use of this information in immunology models. The advent of reinforcement learning with Jarzynski's equality and so-called stiff-spring approximations (Park and Schulten, 2004) to formulate a space of molecular actions using steered MD simulations presents a promising step forward in rapidly modeling at least the 2-state model of the catch bonds as another biophysical descriptor in computational immunology (Choi *et al.*, 2023). A more rigorous consideration of catch bond formation has practical implications for enhancing T cell-based cancer immunotherapies. A recent study showed low-affinity TCRs can be optimized to acquire catch bonding characteristics, allowing for potent activation at relatively weak 3D binding affinities (Zhao *et al.*, 2022). This has the ability to drive a strong antitumor immune response with a lower risk of potentially life-threatening cross-reactivity.

## *Molecular description*

The translation from atomistic to molecular biophysical representation has become popular to allow algorithms to distinguish self versus non-self interactomes. The biophysical representations of glycans underpinning the pathogen entry path offer some stark examples. By employing tools like variational autoencoders, the so-called glycan shield of spike proteins was dissected to detect the role of specific glycan size, orientation, and chemistry (Casalino

et al., 2021). A physical interpretation of the latent spaces was determined from protein-glycan contacts. Subsequently, we engineered the glycan shield based on their contact representation to reduce the infectivity of the NL63 coronavirus by nearly 50% (Chmielewski et al., 2023). This idea of monitoring contacts was also extrapolated to monitor inter-glycan interactions between the cell surface of the influenza virus and those of chicken and human cell surface glycocalyx (Lucas et al., 2021). Again, by translating fluorescence signals into a contact matrix representation, support vector machines were successful in identifying the critical density of glycans that make the H1N1 cells in mammalian cells show a greater binding than when grown in egg cells. Finally, the protein–protein contact matrices also found application in vector design for AstraZeneca and J&J's COVID vaccines, implicating platelet factor proteins in blood clotting side effects of the vaccine candidate (Baker et al., 2021). Altogether, contact matrices can offer a robust biophysical representation, wherein molecular interactions can be classified to be self or non-self.

### Cellular description

Whole-cell models, though scarce, have found applications in computational immunology. A mechanistic, multiscale mathematical model of immunogenicity for therapeutic proteins was formulated by recapitulating key biological mechanisms, including antigen presentation, activation, proliferation, and differentiation of immune cells, secretion of antidrug antibodies, as well as in vivo disposition of antibodies and therapeutic proteins (Chen et al., 2014). The multiscale model structure can be represented by the subcellular, cellular, and whole-body levels. To represent the physiology of MHC-II, a key parameter used in these models involves the number of T-epitope-MHC, in silico T cell epitope prediction and experimental measurements of their MHC-binding affinities, which is scaffolded within a two-compartment drug pharmacokinetics model. Using adalimumab as an example therapeutic protein, the model is able to simulate immune responses against adalimumab in individual subjects and in a population and also provides estimations of immunogenicity incidence and drug exposure reduction that can be validated experimentally (Chen et al., 2014; Handel et al., 2020). Most of the cell models in immunology are agent-based that use the automaton algorithm with specific mechanistic logics or rules. Interestingly these rules show remarkable similarity with classical thermodynamic and kinetic principles, such as landscapes and equations of motion (Koopmans and Youk, 2021). Such models have found applications in CD4+ T cell responses to influenza infections, multiscale mechanistic modeling of human dendritic cells, and have potential applications in dendritic cell-based targeted cell therapies (Wertheim et al., 2021; Aghamiri et al., 2023).

### Macro description

The integration of molecular immunology concepts into macro-level analyses has already demonstrated significant potential in elucidating disease associations. A notable example is the use of patient-specific MHC genotypes to predict disease risk. For instance, large-scale analyses involving 9,176 cancer patients revealed that MHC-I genotypes were predictive of the tumor mutational landscape (Marty et al., 2017). This study found that oncogenic mutations were more likely to occur in regions not presented by the patient's MHC-I molecules, suggesting that gaps in antigen presentation contribute to tumor evolution. Similarly, patients undergoing immune checkpoint blockade therapies have shown improved responses when their MHC-I genotype allows for the presentation of a more diverse array of potential peptides (Chowell et al., 2019). More recently, bio-physical approaches have been applied to link MHC-I genotypes with disease risk and progression (Wilson et al., 2024). Recently, we created a diverse protein ensemble of 5,281 MHC-I protein binding grooves, generating 211,240 structural models, which were subsequently translated into a simplified representation of electrostatic properties (5,281 averaged electrostatic maps). A subset of these maps, those with known MHC-I binding motifs, was used to train an Inception neural network capable of predicting MHC-I binding motifs from electrostatic maps alone. Beyond the ability to perform high-throughput proteome-scale binding predictions, the predicted binding motifs were utilized to construct interaction networks that accurately classified HIV disease progression and immune checkpoint therapy response. At the population level, applications of MHC-I genotype analysis have revealed further insights. A consensus MHC-I prediction model, ensembleMHC, demonstrated that populations enriched for MHC-I alleles capable of strongly binding multiple peptides from SARS-CoV-2 structural proteins exhibited lower mortality rates during the pre-vaccination phase of the COVID-19 pandemic (Wilson et al., 2021). This suggests that MHC-I diversity and peptide-binding capacity at the population level may serve as predictors of disease outcomes in emerging viral threats. These findings highlight some of the promise of MHC genotype-based analysis in both disease risk assessment and therapeutic strategy development. MHC analysis can aid in predicting susceptibility to autoimmune diseases and cancer while also informing vaccine design by optimizing patient antigen selection.

### Outlook: Future inspired by the past of functional representations

Most of the biophysics, including the powerful integrative models we know, is predicated upon the sequence → structure → function → phenotype paradigm. With the maturation of machine learning techniques and the availability of data at various scales, researchers (particularly bioinformaticians) have been trying to bridge gaps between the different tiers of this process, starting from the age-old genotype–type modeling to CASP and AlphaFold's sequence structure up to recent attempts to go from sequence to ensemble. However, physical causality is often missing in the traditional bioinformatics models, thus far sidelining the role of AI-driven advances only to predictions of the forward direction. So, it is high time that we introduce physical ideas to conceive generative models that backmap phenotypes down to an ensemble of structures and sequences. Model representations play a central role in this mapping process. Although the traditional sequence of 3D coordinate structural representations requires an enormous amount of training data and is prone to overfitting, they nonetheless offer the most extensive models. In contrast, the thermodynamic or kinetic representations, using ideas of entropy or committor functions are quite generalizable across application domains but lack the physical interpretability (Mehdi et al., 2024). Loosely, they draw analogies to the plane wave basis set representations that find application in several areas of quantum mechanics (Nagy and Jensen, 2017). However, akin to how quantum mechanics was represented in the molecular systems using the Gaussian-like basis set representations, we posit that biophysical representations offer a segue for representing the deep learning models in the molecular space. To this end, we highlight a number of representations that are either

being used or hold the potential for multiscale applications in computational immunology. Similar to how Gaussian orbitals offer physical interpretation of highly resolved electronic structures (e.g. using the molecular orbital theory), biophysical functions offer interpretability. These functions, such as pocket fields, QM/MM charge density, binding affinity, catch bonding, contact matrices, and molecular electrostatics are deeply rooted in physical theories. These theories (thermodynamic integration, electronic structure theory, equilibrium and non-equilibrium statistical theories, linear response theories, polymer folding, and continuum mechanics) can be projected onto structure and function. Essentially, they offer a physical basis to the loss functions and the latent spaces that enable learning both the data and the context. So, we propose a sustained intellectual effort in this direction.

**Open peer review.** To view the open peer review materials for this article, please visit http://doi.org/10.1017/qrd.2025.7.

**Acknowledgments.** A.S. also acknowledges start-up grants from Arizona State University School of Molecular Sciences and Biodesign Institute's Center for Applied Structural Discovery. A.S. acknowledges funding from the Division of Chemical Sciences, Geosciences, and Biosciences, Office of Basic Energy Sciences, of the U.S. Department of Energy through grants DESC0010575. A.S. acknowledges grant DE-SC0022956 for their support, also from the Department of Energy. This material is based on work supported by the National Defense Education Program (NDEP) for Science, Technology, Engineering, and Mathematics (STEM) Education.

**Financial support.** A.S. was supported by a CAREER award from the NSF (MCB-1942763) and an RO1 grant from the NIH (GM095583).

**Competing interest.** The authors declare no competing interests exist.

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
