## [Reviewer Report]

The manuscript is a well-composed exploration of how biophysical models can enhance traditional computational immunology approaches. The authors emphasize that while sequence-based methods dominate the field, integrating biophysical representations improves efficiency, reduces complexity, and provides deeper insights into immune responses. They discuss advancements in antibody free energy calculations, MHC modeling, and TCR catch bonds, illustrating how these tools address the limitations of sequence-only approaches. Additionally, the manuscript highlights the potential of molecular descriptors, such as contact matrices, for vaccine design and pathogen interaction studies, while connecting biophysical properties to disease progression predictions on a macro scale. The authors advocate for integrating biophysical models to complement existing methods, calling for a shift towards generative models that incorporate physical principles for improved accuracy and relevance.

With minor revisions, the manuscript could be further enhanced:

1) The section on antibody free energy descriptions would benefit from a concluding sentence to connect the findings to the broader theme of biophysical modeling.

2) The sub-sections on MHC and TCR could better summarize the practical implications of these biophysical descriptors.

3) The discussion on MHC-I genotype analysis, while robust, could be strengthened by clarifying how these insights directly influence therapeutic or diagnostic advancements.

---

## [Editor Report]

I have reviewed the authors’ responses for this manuscript (QRBD-2024-0013.R1) and I think that we can accept the revised version without sending the manuscript back to the reviewers. 

Can you please accept this manuscript?